# Differences in Physiological Characteristics of Green Prickly Ash Germplasm Resources in Response to Low-Temperature Stress

**Lin Shi** [1], **Xixi Dong** [1], **Hao Fu** [1,2], **Xingying Chai** [1,2], **Shuqin Bao** [1], **Yun Ren** [1] , **Kai Hu** [1], **Qiang Li** [1,3,*] and **Zexiong Chen** [1,*]

[1] Institute of Special Plants, College of Landscape Architecture and Life Science, Chongqing University of Arts and Sciences, Chongqing 402160, China; cqwushilin@163.com (L.S.); cqwudongxixi@163.com (X.D.); cqwufuhao@163.com (H.F.); cqwuchaixingying@163.com (X.C.); cqwubaoshuqin@163.com (S.B.); reny1989@sina.com (Y.R.); cqwuhukai@163.com (K.H.)
[2] College of Biology and Food Engineering, Chongqing Three Gorges University, Chongqing 404100, China
[3] College of Agronomy and Biotechnology, Southwest University, Chongqing 400715, China
*  Correspondence: liqiangxj@163.com (Q.L.); chenzexiong1979@163.com (Z.C.)

**Abstract:** In this study, we investigated the physiological response to low-temperature stress and comprehensively evaluated the cold resistance of green prickly ash germplasms. One-year-old branches of 23 green prickly ash varieties were used as experimental materials, three branches were selected from each variety, and three replicates of single branches were established. Subsequently, the physiological responses of the 23 green prickly ash germplasm resources to low-temperature stress were analyzed based on the relative conductivity (REC), osmotic adjustment substances, and antioxidant enzyme activities. We found that as the temperature decreased, the REC of each germplasm increased. The most rapid REC increase occurred from 0 to $-10$ °C and continued to gradually gently increase from $-10$ to $-30$ °C. The lethal dose-50 temperature ($LT_{50}$) of the germplasms was between 3.16 and $-12.37$ °C. The top three cold-resistant varieties were TJ, SCHJ, and CJ, and the last three cold-resistant varieties were EWJ, DYSJ, and HYXJ. The results of a correlation analysis revealed a significant correlation between superoxide dismutase (SOD) activity and REC, between REC and free protein, and between SOD activity and catalase. The results of the PCA revealed that the first category includes 5 pepper varieties of CJ, LFJ, and XYTJ with higher cold-resistance score indexes; the second category includes 13 pepper varieties of YQ2 H, WCFZ, and TZJ with appropriate scores; and the third category includes 5 pepper varieties of HYXJ, HNJ, and DYSJ with lower scores. Cluster analysis was performed to classify the cold-resistance physiological indices, and four groups were obtained. The most cold-resistant groups included CJ and LFHJ, whereas the most cold-sensitive groups included HYXJ, DYSJ, and HNJ. Finality, the subordinate function method was used to determine the cold resistance from strong to weak. The varieties with the strongest cold resistance were LFJ, EWJ, and CJ, and the weakest varieties were DYSJ, YQ1H, and HYXJ. The strongest cold-resistant varieties were LFHJ and CJ, followed by DYSJ, YQ1H, and HYXJ. Based on multiple lines of evidence, the most cold-resistant varieties were CJ and LFJ, while the most cold-sensitive varieties were DYSJ and HYXJ. In conclusion, in this study, we elucidated the low-temperature tolerance of different varieties of green prickly ash, and varieties with high cold resistance were selected. These findings provide theoretical guidance and technical support for the screening of cold-resistant green prickly ash germplasms, which will facilitate their introduction to northern China.

**Keywords:** cold stress; peroxidase; electric conductivity; principal component analysis

## 1. Introduction

Prickly ash trees are small perennial trees of the genus *Zanthoxylum*. This group of plants is globally important and known for their medicinal (analgesic with anti-inflammatory properties) and dietary (fruits can be used for spice production) value [1]. In China, prickly ash has become an important tree species in barren mountain management, ecological construction, poverty alleviation, and rural revitalization because of its well-developed root system, drought tolerance, and strong ecological adaptability [2]. In China, two prickly ash species, namely, Red prickly ash (*Zanthoxylum bungeanum* Maxim.) and green prickly ash (*Zanthoxylum armatum* DC.), have been cultivated and consumed [3]. Red prickly ash originated in northern China, with the Taihang Mountains, Yimeng Mountains, northern Shaanxi Plateau, Qinba Mountains, and southern Gansu being the main production areas, whereas green prickly ash originated in Southwest China and is widely cultivated in southern China, with Chongqing, the eastern Sichuan Plateau, and the Yunnan-Guizhou Plateau being the main cultivation areas [4,5]. In recent years, the cultivation of green prickly ash has continuously expanded northward. However, green prickly ash introduced to northern or high-altitude regions frequently experience low-temperature and freezing damage. Therefore, low temperatures have become an important factor limiting the sustainable development of the green prickly ash industry.

The effects of low-temperature stress on fruits and vegetables account for more than 40% of the total effects of various abiotic factors [6]. Under low-temperature stress, plant growth is hindered, fruit yield and quality decline, and plants die in large numbers, resulting in serious losses [7]. Song et al. [8] found that proline can improve resistance to cold damage by protecting protein integrity under low-temperature stress and enhancing the activity of related enzymes. Lin et al. [9] demonstrated that soluble protein can reduce the osmotic potential and improve the water retention capacity of cells. Under low-temperature stress, it can reduce the degree of cell damage by increasing the content of bound water in cells. In a study on the effects of biochar leaching on rice, Yuan Jun [10] found that increasing the content of superoxide dismutase (SOD) and peroxidase (POD) increased the cold resistance of rice. In a simulated low-temperature stress experiment, Wang et al. [11] carried out a low-temperature stress experiment of Qinghai eggplant ginseng leaves, and it was found that with the extension of low-temperature treatment time, the antioxidant enzyme activity of Qinghai eggplant ginseng leaves increased first and then decreased. Sun et al. [12] found that the relative conductivity (REC), osmotic adjustment substances, and protective enzyme activity of coconuts increased with a decrease in temperature. Liu et al. [13] simulated low-temperature stress on walnuts and found that the soluble sugar (SS), soluble protein (SP), and free protein (PRO) contents increased with decreasing temperature. Cao et al. [14] reported that, with the extension of low-temperature stress time, osmotic adjustment substances in palm oil leaves showed an upward trend, and antioxidant enzyme activity initially increased and then decreased. As it is difficult to accurately reflect the cold resistance of plants using a single index, a comprehensive evaluation of cold resistance by combining several main physiological and biochemical indices will increase the accuracy of the results. Liu et al. [15] obtained the lethal dose-50 temperature ($LT_{50}$) by fitting the electrolyte leakage rate with logistic equation, and compared the electrolyte leakage rate and $LT_{50}$ values of three provenances of prickly ash. It was pointed out that YJ pepper had the strongest cold resistance, while DHP and TA1H had weaker cold resistance. Lv et al. [16] used the membership function method to comprehensively analyze the physiological indices of the cold resistance of red prickly ash during cooling and reported that the cold resistance of red prickly ash first increased and then decreased upon exposure to low temperatures in winter.

Previous studies have shown that the cold resistance of Chinese prickly ash germplasm resources varies considerably; however, these studies mostly focused on red prickly ash using a single method and were primarily concentrated in the northern region. To the best of our knowledge, no systematic studies have been conducted to evaluate the cold resistance of green prickly ash in the southern region. Therefore, in this study, 23 green

prickly ash germplasm resources were collected to study the changes in REC, LT$_{50}$, osmotic adjustment substances, and antioxidant enzyme activity under low-temperature stress. The membership function method, principal component analysis (PCA), correlation analysis, and cluster analysis were used to comprehensively evaluate the cold resistance of green prickly ash germplasm resources. The objective of this study was to clarify the tolerance of different varieties of green prickly ash to low temperature and select varieties with strong cold resistance. The results will provide theoretical guidance and technical support for the selection of green prickly ash cultivation sites, thus facilitating the introduction of these trees into northern China.

## 2. Materials and Methods

### 2.1. Overview of the Test Site

This study was conducted in 2022 at the experimental base of the Chongqing University of Arts and Sciences (Yongchuan, Chongqing, China). The germplasm resources were green prickly ash plants that had been growing for more than three years. The test area is characterized by a subtropical monsoon humid climate, with an average annual temperature of 17.7 °C, precipitation of 1015 mm, sunshine of 1218.7 h, and frost-free period of 317 d. The tested soil was purple loam. The nutrients in the 0–30 cm soil layer (at pH 7.01) were as follows: available nitrogen, 36.51 mg/kg; available phosphorus, 105.33 mg/kg; available potassium, 265.72 mg/kg; total nitrogen, 1.13 g/kg; total phosphorus, 15.72 g/kg; total potassium, 1.83 g/kg; and organic matter, 16.21 g/kg.

### 2.2. Testing Material

Twenty-three green prickly ash germplasms were planted on the test base. The origins of each green prickly ash germplasm are listed in Table 1.

**Table 1.** Prickly ash germplasms obtained across China.

| No. | Variety | Sampling Locality | No. | Variety | Sampling Locality |
|---|---|---|---|---|---|
| 1 | RCWC | Rongchang, Chongqing | 13 | HHHJ | Yongshan, Yunnan |
| 2 | WCFZ | Yongchuan, Chongqing, | 14 | LFJ | Yongshan, Yunnan |
| 3 | YL1H | Kunming, Yunnan | 15 | TZJ | Jiangjin, Chongqing |
| 4 | YL2H | Kunming, Yunnan | 16 | JYQ | Jiangjin, Chongqing |
| 5 | YQ1H | Zhaotong, Yunnan | 17 | XJ | Yaan, Sichuan |
| 6 | YQ2H | Zhaotong, Yunnan | 18 | SJ | Jiangjin, Chongqing |
| 7 | HNJ | Honghe, Yunnan | 19 | HYXJ | Qujing, Yunnan |
| 8 | CJ | Yueyang, Hunan | 20 | DYSJ | Honghe, Yunnan |
| 9 | XYTJ | Jiangjin, Chongqing | 21 | QJ | Hanyuan, Sichuan |
| 10 | MLJ | Yingshan, Sichuan | 22 | EWJ | Jiangjin, Chongqing |
| 11 | TJ | Hongya, Sichuan | 23 | SCHJ | Qujing, Yunnan |
| 12 | WCZP | Yongchuan, Chongqing | | | |

### 2.3. Test Method

2.3.1. Experimental Design and Sampling

In this experiment, five trees with similar growth vigor were selected from the green prickly ash test base of Chongqing University of Arts and Sciences in May 2022, and 1-year-old branches (middle stems) with a length of 25–30 cm and diameter of 0.8–1.0 cm were randomly cut from the middle of each tree and immediately transported back to the laboratory. The branches were washed with pure water, and the branches were sealed with paraffin to prevent water loss. The sealed branches were placed in the refrigerator to simulate low-temperature stress treatment. In this experiment, 0 °C was used as the control, and −10, −20, and −30 °C temperature treatments were established. Three branches of each green prickly ash germplasm were treated under different low-temperature stress conditions, to increase the reliability of the experiment. The refrigerator cooling rate was set to 5 °C/h, to gradually lower the temperature; the set temperature was reached after low-

temperature treatment for 12 h, after which the temperature was raised to 5 °C and thawed for 12 h. Subsequently, we collected the middle sections of the branches to determine the physiological and biochemical indices. Each treatment was repeated three times.

2.3.2. Determination of Physiological and Biochemical Indexes

REC (relative conductivity) was determined according to the method described by Lu et al. [17], and the $LT_{50}$ was calculated. The contents of SS (soluble sugar), SP (soluble protein), and PRO (free protein) were determined using anthrone colorimetry, Coomassie Brilliant Blue, and acid ninhydrin methods according to Liu et al. [18]. The activities of SOD (superoxide dismutase), POD (peroxidase), and catalase (CAT) were determined using nitroblue tetrazolium, guaiacol, and ultraviolet absorption methods, as described by Huang et al. [19].

*2.4. Data Statistics and Analysis*

Data were sorted using Excel (2019, Microsoft Corporation, Redmond, DC, USA). One-way analysis of variance was performed for significance testing, and Tukey's method was used to test for statistical differences. Chiplot mapping software and SPSS 19.0 were used for the statistical analyses of the data.

The logistic equation was fitted with the treatment temperature x and relative conductivity Y as shown in Formula (1):

$$Y = \frac{k}{1 + ae^{-bx}} \tag{1}$$

where k = 100 represents the saturated capacity of the cell injury rate. In Formula (1), the parameters a and b are used to fit the data. The parameter a controls the slope of the curve, and b controls the shape of the curve. In Formula (2), lna represents the natural logarithm of parameter a. Parameters a and b and the fitting degree $R^2$ were determined, and the $LT_{50}$ was obtained according to Formula (2):

$$LT_{50} = \frac{\ln a}{b} \tag{2}$$

The comprehensive evaluation formula of the membership function method is shown below. The index that positively correlated with cold resistance was calculated using Equation (3):

$$f(x_{ij}) = \frac{X_{ij} - X_{jmin}}{X_{jmax} - X_{jmin}} \tag{3}$$

where $f(x_{ij})$ is the membership function value of index j for variety i, $X_{ij}$ is the measured value, $X_{jmin}$ is the minimum value of the index j, and $X_{jmax}$ is the maximum value of the index j. The index that negatively correlated with cold resistance was calculated using Equation (4):

$$f(x_{ij}) = 1 - \frac{X_{ij} - X_{jmin}}{X_{jmax} - X_{jmin}} \tag{4}$$

*2.5. Principal Component Analysis*

First, KMO and Bartlett's sphericity tests were performed on the standardized data to examine the suitability of principal component extraction. Then, the eigenvalues and variance contribution rates of each component were analyzed. Finally, the variance contribution rate corresponding to the extracted principal components was used as the weight, and the weighted summation method was used to calculate the comprehensive score of the membership function of different peppers under low-temperature stress. The calculation formula is as follows:

$$F = A_1F_1 + A_2F_2 + A_3F_4 \ldots \ldots + A_nF_n$$

In the formula, F is the comprehensive score, and $F_n$ is the score of the nth principal component; An is the variance contribution rate of the nth principal component.

### 2.6. Cluster Analysis

Clustering analysis is a kind of statistical method to classify samples or variables. The data to be analyzed are the similarity or dissimilarity between objects. The similarity (dissimilarity) of these data is measured by the 'distance' between objects. The objects that are close to each other are classified into one class, and the objects that are far away are regarded as different classes. DPS was used for intra-group cluster analysis mapping.

## 3. Results

### 3.1. REC and $LT_{50}$ of Green Prickly Ash Germplasm under Low-Temperature Stress

Under low-temperature stress, the REC of the green prickly ash germplasm branches differed significantly ($p < 0.05$). The REC of the branches of each green prickly ash germplasm increased significantly with decreasing temperatures (Figure 1, Table S1). Under the 0 °C treatment, the average REC of the 23 green prickly ash germplasms was 27.09%, among which the REC values of QJ and TJ were the highest and lowest at 48.68% and 8.48%, respectively. Thus, the difference between the highest and lowest REC values of these varieties was 40.20%. Under the −10 °C treatment, the average REC was 67.39%, among which the REC values of HYXJ and CJ were the highest and lowest at 86.63% and 54.69%, respectively. Thus, the difference between the highest and lowest REC values of these varieties was 31.97%. Under the −20 °C treatment, the average REC was 82.08%, among which the REC values of RCWC and CJ were the highest and lowest at 92.26% and 72.55%, respectively. Thus, the difference between the highest and the lowest REC values of these varieties was 19.71%. Under the −30 °C treatment, the REC was 89.90% on average, among which the REC values of XYTJ and MLJ were the highest and lowest at 94.03% and 82.46%, respectively. Thus, the difference between the highest and the lowest REC values of these varieties was 11.57%. When the temperature decreased from 0 to −10 °C, the REC of each green prickly ash germplasm increased by 40.31% on average, among which the REC values of XYTJ and DYSJ showed the highest and lowest increases at 53.24% and 21.34%, respectively. When the temperature decreased from −10 to −20 °C, the upward REC trend slowed down, and the average increased by 14.69%. Among them, YL1H and HYXJ showed the highest and lowest increases at 27.03% and 3.96%, respectively. When the temperature decreased from −20 to −30 °C, the REC of green prickly ash continued to increase but the range of increase further decreased. The average REC only increased by 7.82%, among which YQ2H increased to the greatest extent (15.64%), HYXJ increased the least (0.84%), and RCWC and MLJ decreased by 1.96% and 2.88%, respectively.

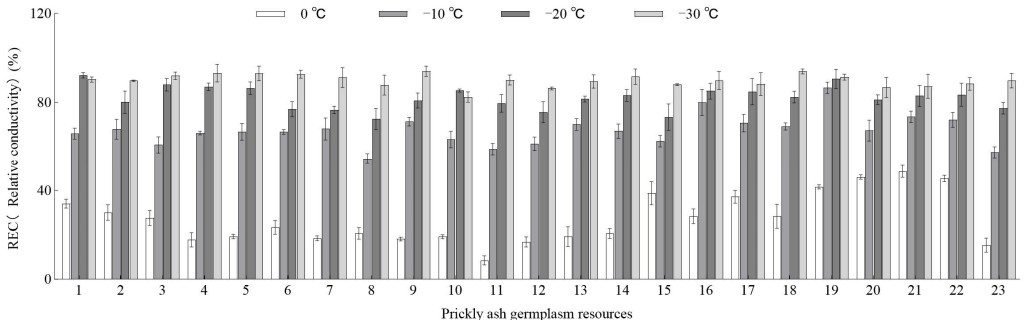

**Figure 1.** Changes in relative conductivity (REC) content in green prickly ash germplasm from different origins under different low-temperature stress. Note: The digital details are shown in Table 1, the same below.

The fitting results of the logistic equation for the REC of green prickly ash germplasm branches are shown in Table 2. The correlation coefficients were between 0.92 and 0.99,

indicating that the degree of fit of the equation was good and the results of calculating $LT_{50}$ were reliable and accurate. The average $LT_{50}$ of the 23 green prickly ash germplasms was $-6.17\ ^\circ$C. The $LT_{50}$ values of TJ, SCHJ, CJ, and WCZP without thorns were $-12.37\ ^\circ$C, $-11.02\ ^\circ$C, $-10.84\ ^\circ$C, and $-10.81\ ^\circ$C, respectively. The $LT_{50}$ values of HYXJ, QJ, and EWJ were $3.16\ ^\circ$C, $2.24\ ^\circ$C, and $0.13\ ^\circ$C, respectively. The lower the $LT_{50}$ of the plants, the stronger the cold resistance, and vice versa, indicating that the cold resistance of TJ, SCHJ, and CJ was stronger, while the cold resistance of HYXJ, DYSJ, and EWJ was weaker.

**Table 2.** Logistic equation fitting results of the relative conductivity (REC) of branches of different varieties of green prickly ash under low-temperature stress.

| Variety | Logistic Equation | $LT_{50}$ (°C) | Degree of Fitting |
|---|---|---|---|
| RCWC | $y = 100/(1 + 1.49e^{0.11x})$ | $-3.78$ | 0.98 * |
| WCFZ | $y = 100/(1 + 1.80e^{0.10x})$ | $-6.04$ | 0.98 * |
| YL1H | $y = 100/(1 + 2.20e^{0.12x})$ | $-6.69$ | 0.99 ** |
| YL2H | $y = 100/(1 + 3.12e^{0.14x})$ | $-8.31$ | 0.98 * |
| YQ1H | $y = 100/(1 + 2.88e^{0.13x})$ | $-7.99$ | 0.98 * |
| YQ2H | $y = 100/(1 + 2.56e^{0.12x})$ | $-8.04$ | 0.97 * |
| HNJ | $y = 100/(1 + 2.96e^{0.12x})$ | $-9.12$ | 0.95 * |
| CJ | $y = 100/(1 + 3.20e^{0.11x})$ | $-10.84$ | 0.99 * |
| XYTJ | $y = 100/(1 + 3.00e^{0.13x})$ | $-8.24$ | 0.96 * |
| MLJ | $y = 100/(1 + 2.52e^{0.10x})$ | $-9.07$ | 0.94 * |
| TJ | $y = 100/(1 + 6.20e^{0.15x})$ | $-12.37$ | 0.96 * |
| WCZP | $y = 100/(1 + 3.28e^{0.11x})$ | $-10.81$ | 0.96 * |
| HHHJ | $y = 100/(1 + 2.57e^{0.11x})$ | $-8.31$ | 0.95 * |
| LFJ | $y = 100/(1 + 2.66e^{0.12x})$ | $-8.08$ | 0.97 * |
| TZJ | $y = 100/(1 + 1.51e^{0.08x})$ | $-5.22$ | 0.99 ** |
| JYQ | $y = 100/(1 + 1.43e^{0.10x})$ | $-3.67$ | 0.92 * |
| XJ | $y = 100/(1 + 1.28e^{0.09x})$ | $-2.92$ | 0.97 * |
| SJ | $y = 100/(1 + 2.05e^{0.12x})$ | $-6.11$ | 0.98 * |
| HYXJ | $y = 100/(1 + 0.76e^{0.09x})$ | 3.16 | 0.92 * |
| DYSJ | $y = 100/(1 + 0.87e^{0.07x})$ | 2.24 | 0.97 * |
| QJ | $y = 100/(1 + 1.05e^{0.07x})$ | $-0.72$ | 0.99 ** |
| EWJ | $y = 100/(1 + 0.99e^{0.07x})$ | 0.13 | 0.98 * |
| SCHJ | $y = 100/(1 + 4.00e^{0.13x})$ | $-11.02$ | 0.98 * |

Note: "**" indicates an extremely significant difference at 1% ($p < 0.01$) and "*" indicates a significant difference at 5% ($p < 0.05$).

### 3.2. Effect of Low-Temperature Stress on Osmotic Adjustment Substances of Green Prickly Ash
3.2.1. Effect of Low-Temperature Stress on the SS Content of Green Prickly Ash

Under low-temperature stress, a significant difference was observed in SS content among the green prickly ash germplasms ($p < 0.05$). With decreasing temperature, the SS content of the green prickly ash germplasm initially increased and then decreased (Figure 2, Table S2). Under the 0 °C treatment, the average SS content was 287.14 μg/mL, with QJ having the highest SS content, reaching 419.37 μg/mL, and SCHJ having the lowest at 146.03 μg/mL. Thus, the difference between the highest and lowest values was 273.14 μg/mL. Under the $-10\ ^\circ$C treatment, the average SS content was 319.05 μg/mL, YL2H had the highest SS content at 447.43 μg/mL, and RCWC had the lowest at 167.77 μg/mL. Thus, the difference between the highest and lowest values was 279.66 μg/mL. Under the $-20\ ^\circ$C treatment, the average SS content was 306.94 μg/mL, with WCZP having the highest content of 448.47 μg/mL and YL1H having the lowest content of 156.10 μg/mL. Thus, the difference between the highest and lowest values was 292.37 μg/mL. When the temperature decreased to $-30\ ^\circ$C, the average SS content was 254.31 μg/mL, with LFJ having the highest content at 420.37 μg/mL, and WCZP having the lowest content at 138.97 μg/mL. Thus, the difference between the highest and lowest contents was 147.23 μg/mL. When the temperature decreased from 0 to $-10\ ^\circ$C, most of the varieties showed an increase, and a few showed a decrease, and the average SS changed by 36.98%. Among them, the SS content of MLJ changed the most at 127.36%, while

that of TZJ changed the least at only 4.20%. When the temperature decreased from −10 to −20 °C, most of the varieties showed an increase, and a few showed a decrease, and the average change in each green prickly ash germplasm was 37.95%. Among them, the change in RCWC without thorns was the largest at up to 134.99% and the change in TZJ was the smallest at only up to 3.62%. When the temperature decreased from −20 to −30 °C, most of the varieties showed a decrease, and a few showed an increase, and the average change in green prickly ash germplasm was 34.29%. Among them, the change in SJ was the largest at up to 73.93% and the change in XJ was the smallest at 3.84%.

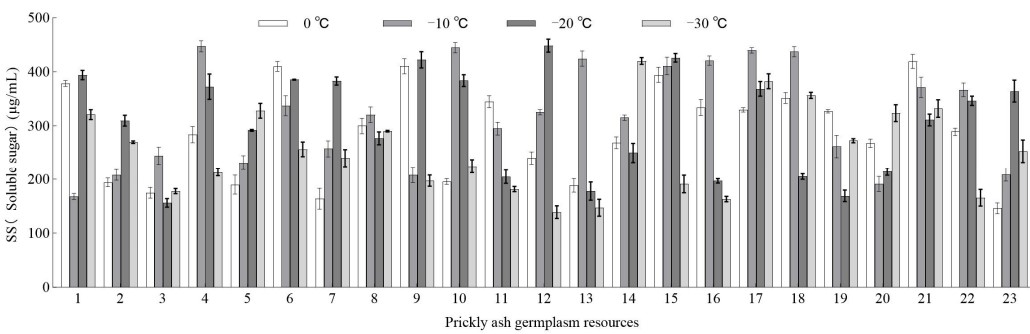

**Figure 2.** Changes in the soluble sugar (SS) content of the green prickly ash germplasms from different origins under different low-temperature stress.

### 3.2.2. Effect of Low-Temperature Stress on the SP Content of Green Prickly Ash

Under low-temperature stress, the SP content of green prickly ash germplasms presented significant differences ($p < 0.05$). With decreasing temperature, the SP content of the green prickly ash germplasm initially increased, then decreased, and finally increased (Figure 3, Table S3). Under the 0 °C treatment, the average SP content of the 23 green prickly ash germplasms was 175.29 ng/mL, among which HYXJ had the highest SP content at 224.50 ng/mL and YQ1H had the lowest at 132.00 ng/mL. Thus, the difference between the highest and lowest values was 92.50 ng/mL. Under the −10 °C treatment, the average SP content was 191.71 ng/mL, among which XYTJ had the highest and RCWC had the lowest SP content at 233.30 ng/mL and 132.30 ng/mL, respectively. Thus, the difference between the highest and lowest values was 101.00 ng/mL. Under the −20 °C treatment, the average SP content was 173.38 ng/mL, among which WCFZ had the highest and LFJ had the lowest SP content at 237.50 ng/mL and 130.33 ng/mL, respectively. Thus, the difference between the highest and lowest values was 107.17 ng/mL. When the temperature decreased to −30 °C, the average SP content was 188.97 ng/mL, among which CJ had the highest and SJ had the lowest at 231.97 ng/mL and 141.70 ng/mL, respectively. Thus, the difference between the highest and lowest values was 90.27 ng/mL. When the temperature decreased from 0 to −10 °C, the SP content of each green prickly ash germplasm changed by 22.62% on average, among which HHHJ changed the most to 50.61% and EWJ changed the least to 2.75%. When the temperature decreased from −10 to −20 °C, the average change in SP content was 24.53% in green prickly ash germplasms, with the largest increase of 52.11% in WCFZ germplasms without thorns and the smallest increase of 2.39% in YQ1H. When the temperature decreased from −20 to −30 °C, the SP content of each green prickly ash germplasm changed by 20.99% on average, with the largest increase of 67.83% in LFJ and the smallest increase of 1.65% in RCWC.

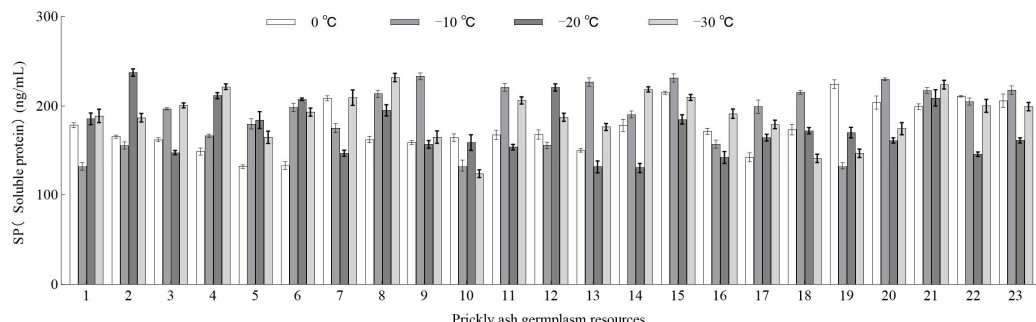

**Figure 3.** Changes in the soluble protein (SP) content of the green prickly ash germplasms from different origins under different low-temperature stress.

### 3.2.3. Effect of Low-Temperature Stress on the PRO Content of Green Prickly Ash

Under low-temperature stress, the Pro content in the branches of the green prickly ash germplasm differed significantly ($p < 0.05$). With decreasing temperature, the Pro content in the branches of the green prickly ash germplasm first decreased and then increased (Figure 4, Table S4). At 0 °C, the average PRO content of the 23 green prickly ash germplasms was 6.30 ng/mL, among which LFJ had the highest at 8.20 ng/mL and WCZP had the lowest at only 3.65 ng/mL. Thus, the difference between the highest and lowest values was 4.55 ng/mL. Under the −10 °C treatment, the average Pro content was 5.56 ng/mL, among which YL1H had the highest and TZJ had the lowest content at 7.90 ng/mL and 3.33 ng/mL, respectively. Thus, the difference between the highest and lowest values was 4.57 ng/mL. Under the −20 °C treatment, the average Pro content was 5.25 ng/mL, among which JYQ had the highest and RCWC had the lowest content at 7.76 ng/mL and 3.52 ng/mL, respectively. Thus, the difference between the highest and lowest values was 4.24 ng/mL. When the temperature decreased to −30 °C, the average Pro content was 6.11 ng/mL, with the highest in XJ and the lowest in YQ1H at 8.33 ng/mL and 3.77 ng/mL, respectively. Thus, the difference between the highest and lowest was 4.56 ng/mL. When the temperature decreased from 0 to −10 °C, the Pro content of each green prickly ash germplasm changed by 27.20% on average, with YL2H changing the most by 78.24% and HHHJ changing the least by 1.27%. When the temperature decreased from −10 to −20 °C, the change in the Pro content of green prickly ash slowed down, and the average change in each green prickly ash germplasm was 19.32%. YL1H changed the most by up by 52.52%, and XJ changed the least and decreased by 0.22%. When the temperature decreased from −20 to −30 °C, the Pro content of green prickly ash increased again. The average change in the green prickly ash germplasm was 27.93%. The change in TJ was the largest and increased by 72.55%, whereas the change in YQ2H was the smallest and decreased by 0.23%.

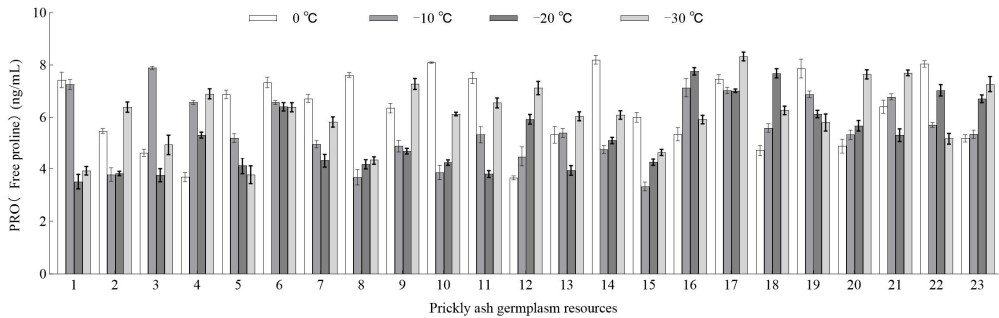

**Figure 4.** Changes in the free protein (PRO) content of the green prickly ash germplasms from different origins under different low-temperature stress.

### 3.3. Effect of Low-Temperature Stress on Antioxidant Enzyme System of Green Prickly Ash

3.3.1. Effect of Low-Temperature Stress on SOD Activity of Green Prickly Ash

Under low-temperature stress, the differences in SOD activity among the green prickly ash germplasms were significant ($p < 0.05$). With decreasing temperature, the SOD activity of the green prickly ash germplasm first decreased, then increased, and finally decreased (Figure 5, Table S5). Under the 0 °C treatment, the average SOD activity was 110.48 U/mL, among which YQ2H had the highest activity at 185.88 U/mL and XJ had the lowest activity at 35.88 U/mL. Thus, the difference between the highest and lowest values was 150.00 U/mL. Under the −10 °C treatment, the average SOD activity was 93.78 U/mL, among which MLJ had the highest and SJ had the lowest activity at 173.00 U/mL and 32.93 U/mL, respectively. Thus, the difference between the highest and lowest values was 140.07 U/mL. Under the treatment of −20 °C, the average SOD activity was 117.99 U/mL, among which WCZP had the highest and RCWC had the lowest at 182.99 U/mL and 43.43 U/mL, respectively. Thus, the difference between the highest and lowest values was 139.56 U/mL. When the temperature decreased to −30 °C, the average SOD activity was 95.15 U/mL, the highest was still WCZP and YQ1H had the lowest activity at 167.87 U/mL and 37.16 U/mL, respectively. Thus, the difference between the highest and lowest values was 130.71 U/mL. When the temperature decreased from 0 to −10 °C, the average change in each green prickly ash germplasm was 61.90%, among which YL1H showed the largest change and increased by 266.51% and JYQ showed the smallest change and increased by 7.16%. When the temperature decreased from −10 to −20 °C, the change in the SOD activity of green prickly ash was still increasing, and the average change in the green prickly ash germplasm was 73.85%. Among them, YL2H changed the most and increased by 300.56% and YL1H changed the least and increased by 1.64%. When the temperature decreased from −20 to −30 °C, the change in SOD activity of green prickly ash slowed down, and the average change in each green prickly ash germplasm was 33.32%. Among them, the change in EWJ was the largest and increased by 97.39%, while HYXJ was the smallest and increased by only 5.73%.

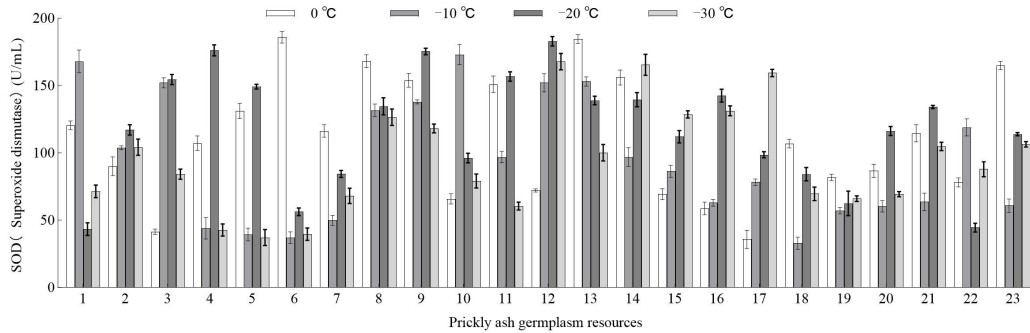

**Figure 5.** Changes in the superoxide dismutase (SOD) content of the green prickly ash germplasms from different origins under different low-temperature stress.

3.3.2. Effect of Low-Temperature Stress on the POD Activity of Green Prickly Ash

Under low-temperature stress, the POD activity of the branches of the green prickly ash germplasm differed significantly ($p < 0.05$). With decreasing temperature, the POD activity of the branches of each green prickly ash germplasm first decreased and then increased (Figure 6, Table S6). Under the 0 °C treatment, the average POD activity of the 23 green prickly ash germplasms was 320.80 U/L, among which JYQ had the highest and TZJ had the lowest activity at 401.27 U/L and 234.30 U/L, respectively. Thus, the difference between the highest and lowest values was 166.97 U/L. Under the −10 °C treatment, the average POD activity was 303.90 U/L, among which CJ had the highest and HHHJ had the lowest activity at 401.93 U/L and 218.93 U/L, respectively. Thus, the difference between the highest and lowest values was 183.00 U/L. Under the −20 °C treatment, the average POD activity was 305.87 U/L, among which YL2H had the highest and YL1H had the

lowest activity at 405.07 U/L and 214.07 U/L, respectively. Thus, the difference between the highest and lowest values was 191.00 U/L. When the temperature decreased to −30 °C, the average POD activity was 310.68 U/L, among which the POD activity of YL2H was still the highest at 399.70 U/L and that of DYSJ had the lowest at 231.83 U/L. Thus, the difference between the highest and lowest values was 167.87 U/L. When the temperature decreased from 0 to −10 °C, the average change in POD activity of each green prickly ash germplasm was 19.11%. Among them, TZJ changed the most and increased by 58.97%, while YL2H changed the least and decreased by 2.11%. When the temperature decreased from −10 to −20 °C, the average change in POD activity was 23.96%. Among them, the change in the POD activity of WCZP was the largest and increased by 66.93% while that of WCFZ changed the least and decreased by 0.54%. When the temperature decreased from −20 to −30 °C, the change in the POD activity of green prickly ash slowed down, and the average change in each green prickly ash germplasm was 15.63%. Among them, YQ1H changed the most and increased by 37.89%, while TZJ changed the least and decreased by 1.18%.

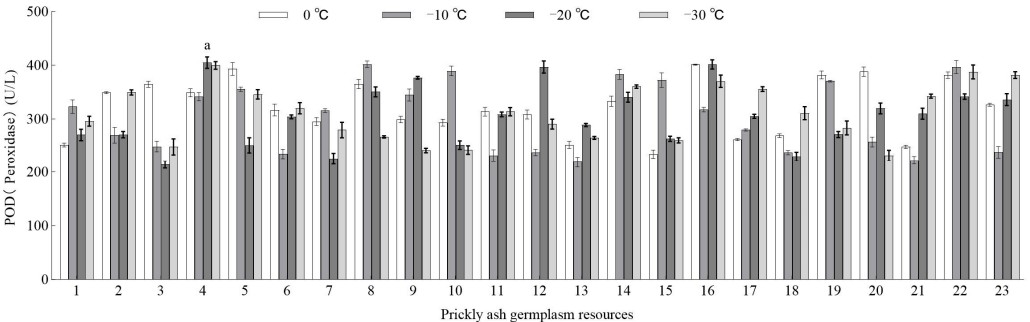

**Figure 6.** Changes in the peroxidase (POD) content of green prickly ash germplasms from different origins under different low-temperature stress.

### 3.3.3. Effect of Low-Temperature Stress on CAT Activity of Green Prickly Ash

Under low-temperature stress, a significant difference was observed in CAT activity among green prickly ash germplasms ($p < 0.05$). With decreasing temperature, the CAT activity of the green prickly ash germplasm initially increased and then decreased (Figure 7, Table S7). Under the 0 °C treatment, the average CAT activity of the 23 green prickly ash germplasms was 60.96 U/mL, among which XJ had the highest activity at 86.09 U/mL and HNJ had the lowest activity at 39.03 U/mL. Thus, the difference between the highest and lowest values was 47.06 U/mL. Under the −10 °C treatment, the average CAT activity was 62.65 U/mL, among which EWJ had the highest activity and YQ2H had the lowest activity at 85.75 U/mL and 44.64 U/mL, respectively. Thus, the difference between the highest and lowest values was 41.11 U/mL. Under the −20 °C treatment, the average CAT activity was 67.14 U/mL, among which CJ had the highest at 89.39 U/mL and DYSJ had the lowest activity at 39.91 U/mL. Thus, the difference between the highest and lowest values was 49.48 U/mL. When the temperature was reduced to −30 °C, the average CAT activity was 61.35 U/mL, of which HHHJ had the highest and QJ had the lowest activity at 86.89 U/mL and 39.72 U/mL, respectively. Thus, the difference between the highest and lowest values was 47.17 U/mL. When the temperature decreased from 0 to −10 °C, the average change in the CAT activity of each green prickly ash germplasm was 23.68%. Among them, HNJ changed the most and increased by 74.17% while YQ1H changed the least and increased by 3.05%. When the temperature decreased from −10 to −20 °C, the change in CAT activity in green prickly ash continued to intensify, and the average change in each green prickly ash germplasm was 30.21%. Among them, the change in CJ was the largest and increased by 77.90%, while the change in EWJ was the smallest and decreased by 1.60%. When the temperature decreased from −20 to −30 °C, the average change in CAT activity of each green prickly ash germplasm was 28.59%. Among them, the change in

DYSJ was the largest and increased by 81.32%, while the change in EWJ was the smallest and decreased by 1.40%.

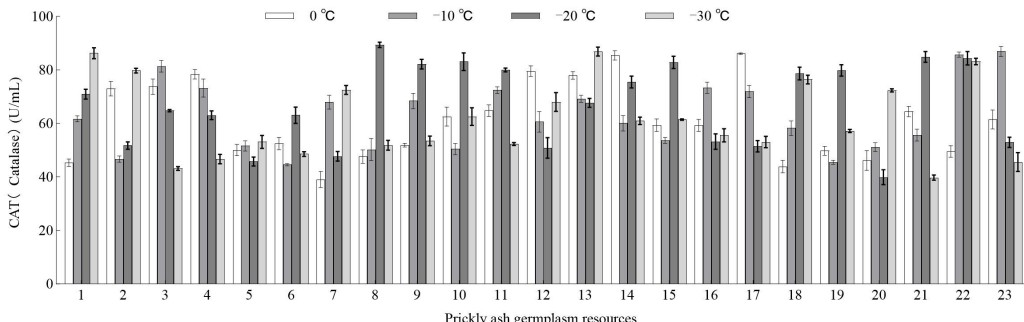

**Figure 7.** Changes in the catalase (CAT) content of the green prickly ash germplasms from different origins under different low-temperature stress.

*3.4. Correlation Analysis of the Subordinate Function Values of Cold Resistance Indexes of Green Prickly Ash*

Correlation analysis of the 23 green pepper germplasm resources (Figure 8) revealed a correlation between each index. Specifically, a significant negative correlation was observed between SOD activity and REC, with a correlation coefficient of −0.56. There was a significant positive correlation between REC and PRO and between SOD and CAT activity, with correlation coefficients of 0.5 and 0.43, respectively. These results indicate the overlap of the cold-resistance-related insights brought forth by multiple indicators. However, the patterns were exhibited by the respective indices varied (i.e., across different germplasm resources), indicating differing roles played by these indices in the response to cold stress. Therefore, to eliminate the bias of a single cold-resistance indicator, a comprehensive analysis is required to assess the cold resistance of green pepper germplasm resources. For the comprehensive evaluation, REC, SS, SP, PRO, SOD, POD, and CAT were selected as reference indicators and evaluated using PCA.

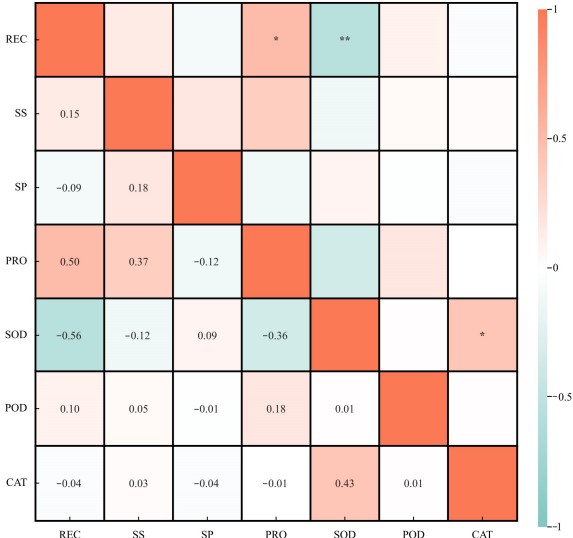

**Figure 8.** Correlation analysis of the subordinate function values of cold-resistance indexes of green prickly ash. Note: "*" indicates a significant difference at $p < 0.05$, "**" indicates an extremely significant difference at $p < 0.01$.

The cold-resistance coefficients of the seven indicators were subjected to PCA using SPSS software (19.0, IBM Corporation, Armonk, NY, USA). Three principal components were extracted based on eigenvalues greater than one. The three principal components

contributed variances of 30.897%, 18.179%, and 16.305% respectively, with a cumulative variance contribution of 65.381% (Table 3).

**Table 3.** Main component analysis of the physiological indexes.

| Ingredient | Principal Component Eigenvalues | | |
|---|---|---|---|
| | Characteristic Value λ | Proportion of Factors % | Total Percentage % |
| 1 | 2.16 | 30.90 | 30.90 |
| 2 | 1.27 | 18.18 | 49.08 |
| 3 | 1.14 | 16.31 | 65.38 |

Using the correlation analysis method of mathematical statistics, the eigenvector (weight coefficient) of the three principal components (W1, W2, and W3) was obtained by calculating the eigenvalue and factor-load matrix of each index (Table 4).

**Table 4.** Weighted coefficient of each physiological index under the three main components.

| Physiological Index | Weight Coefficient of Each Index under Each Principal Component | | |
|---|---|---|---|
| | W1 | W2 | W3 |
| REC | 0.80 | 0.02 | −0.16 |
| SS | 0.41 | 0.56 | 0.45 |
| SP | 0.13 | 0.23 | 0.83 |
| PRO | 0.77 | 0.34 | −0.13 |
| SOD | −0.79 | 0.43 | −0.10 |
| POD | 0.19 | 0.37 | −0.24 |
| CAT | −0.31 | 0.69 | −0.39 |

As shown in Figure 9, the cumulative contribution rate of the two principal components PC1 (46.48%) and PC2 (21.74%) reached 68.22%, indicating that the two factors can be used to replace the original data to evaluate the membership function values of cold resistance indexes of the 23 green prickly ash varieties. The first category is the 5 pepper varieties with higher cold-resistance scores, including CJ, LFJ, and XYTJ. These 5 varieties had higher SOD values, indicating that the SOD value occupies an important position in PC1. The second category was the 13 pepper varieties with appropriate scores, including YQ2H, WCFZ, and TZJ. The third category was the 5 pepper varieties with lower scores, including HYXJ, HNJ, and DYSJ. These five varieties had higher SP values, indicating that the SP value occupies an important position in PC3.

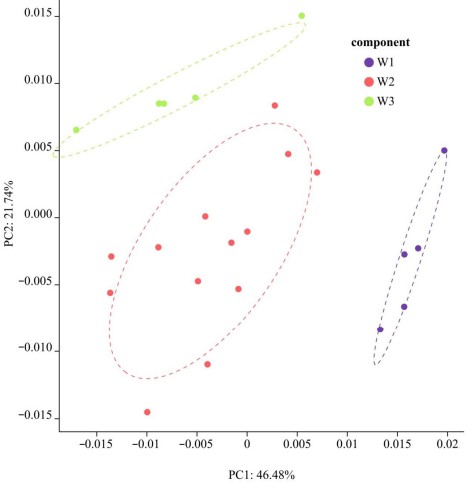

**Figure 9.** Principal component analysis of the membership function value of cold resistance index of green prickly ash.

Calculation of the membership function values of the seven indicators; the degree of cold resistance of different green pepper varieties was inconsistent (Figure 8). Using an index to evaluate the cold resistance of green prickly ash will lead to different results; therefore, the comprehensive evaluation method is used to evaluate the cold resistance of plants.

### 3.5. Comprehensive Evaluation of Cold Resistance of Green Prickly Ash Germplasm Resources

Owing to the different units and properties of each index, the membership function method was used to evaluate the cold-resistance-related indices comprehensively. First, the membership function value of each index for each variety was calculated according to the membership function formula, and then the average membership value of each variety was calculated. The greater the average membership value, the stronger the cold resistance of the variety. The average membership value of green prickly ash was between 0.33 and 0.69 (Table 5), and the cold resistance of different varieties differed considerably. According to the average membership value, the cold resistance of green prickly ash was divided into four grades: Grade 1 included six varieties, LFJ, CJ, EWJ, WCZP, YL2H, and XYTJ, which had average membership values greater than 0.60 and were strongly cold-resistant varieties; Grade 2 included six varieties, XJ, SCHJ, TJ, QJ, TZJ, and JYQ, with an average membership value between 0.50 and 0.60, which were medium cold-resistant varieties. Grade 3 included six varieties, HHHJ, YQ2H, MLJ, RCWC, WCFZ, and SJ, with the average membership degree between 0.40 and 0.50, which were weakly cold-resistant varieties. Grade 4 included five varieties, HNJ, YL1H, DYSJ, YQ1H, and HYXJ, with an average membership degree of less than 0.40. These were the cold-sensitive rice varieties.

**Table 5.** Subordinate function values of cold resistance indexes of green prickly ash.

| Variety | Subordinate Function Values | | | | | | | | Ranking |
|---------|------|------|------|------|------|------|------|---------|---------|
| | REC | SS | SP | PRO | SOD | POD | CAT | Average | |
| RCWC | 0.37 | 0.66 | 0.36 | 0.33 | 0.43 | 0.24 | 0.62 | 0.43 | 16 |
| WCFZ | 0.57 | 0.30 | 0.57 | 0.11 | 0.47 | 0.44 | 0.49 | 0.42 | 17 |
| YL1H | 0.56 | 0.00 | 0.44 | 0.26 | 0.52 | 0.11 | 0.61 | 0.36 | 20 |
| YL2H | 0.62 | 0.73 | 0.58 | 0.37 | 0.32 | 0.97 | 0.59 | 0.60 | 5 |
| YQ1H | 0.60 | 0.38 | 0.28 | 0.15 | 0.28 | 0.66 | 0.00 | 0.34 | 22 |
| YQ2H | 0.68 | 0.83 | 0.53 | 0.73 | 0.16 | 0.31 | 0.08 | 0.47 | 14 |
| HNJ | 0.75 | 0.38 | 0.55 | 0.31 | 0.16 | 0.19 | 0.26 | 0.37 | 19 |
| CJ | 1.00 | 0.57 | 0.77 | 0.13 | 0.92 | 0.75 | 0.38 | 0.65 | 2 |
| XYTJ | 0.61 | 0.63 | 0.47 | 0.43 | 1.00 | 0.49 | 0.54 | 0.60 | 6 |
| MLJ | 0.80 | 0.65 | 0.00 | 0.35 | 0.46 | 0.31 | 0.57 | 0.45 | 15 |
| TJ | 0.98 | 0.36 | 0.58 | 0.43 | 0.62 | 0.30 | 0.67 | 0.56 | 9 |
| WCZP | 0.94 | 0.52 | 0.53 | 0.25 | 0.97 | 0.43 | 0.57 | 0.60 | 4 |
| HHHJ | 0.67 | 0.24 | 0.36 | 0.21 | 0.97 | 0.00 | 0.99 | 0.49 | 13 |
| LFJ | 0.64 | 0.65 | 0.48 | 0.51 | 0.91 | 0.81 | 0.80 | 0.69 | 1 |
| TZJ | 0.64 | 0.87 | 0.90 | 0.00 | 0.41 | 0.22 | 0.55 | 0.51 | 11 |
| JYQ | 0.36 | 0.47 | 0.29 | 0.68 | 0.40 | 0.96 | 0.40 | 0.51 | 12 |
| XJ | 0.39 | 1.00 | 0.37 | 1.00 | 0.33 | 0.37 | 0.61 | 0.58 | 7 |
| SJ | 0.49 | 0.78 | 0.42 | 0.52 | 0.08 | 0.05 | 0.55 | 0.41 | 18 |
| HYXJ | 0.00 | 0.36 | 0.33 | 0.73 | 0.00 | 0.58 | 0.31 | 0.33 | 23 |
| DYSJ | 0.39 | 0.32 | 0.66 | 0.46 | 0.21 | 0.36 | 0.09 | 0.35 | 21 |
| QJ | 0.24 | 0.89 | 1.00 | 0.69 | 0.47 | 0.20 | 0.43 | 0.56 | 10 |
| EWJ | 0.28 | 0.54 | 0.63 | 0.67 | 0.20 | 1.00 | 1.00 | 0.62 | 3 |
| SCHJ | 0.94 | 0.29 | 0.71 | 0.54 | 0.56 | 0.53 | 0.45 | 0.57 | 8 |

The top three average membership function values for the 23 green prickly ash germplasms were CJ, LFJ, and EWJ, which were 0.69, 0.65, and 0.62, respectively. The last three were DYSJ, YQ1H, and HYXJ, with values of 0.35, 0.34, and 0.33, respectively. The higher the average membership function value, the stronger the cold resistance, and

vice versa. This indicates that the cold resistance of LFJ, CJ, and EWJ was stronger, whereas the cold resistances of DYSJ, YQ1H, and HYXJ were weaker.

### 3.6. Cluster Analysis of the Physiological and Biochemical Parameters of Different Green Prickly Ash Varieties

The physiological and biochemical parameters related to cold resistance in the 23 green prickly ash germplasms were analyzed using cluster analysis and divided into four groups at five Euclidean distances (Figure 10).

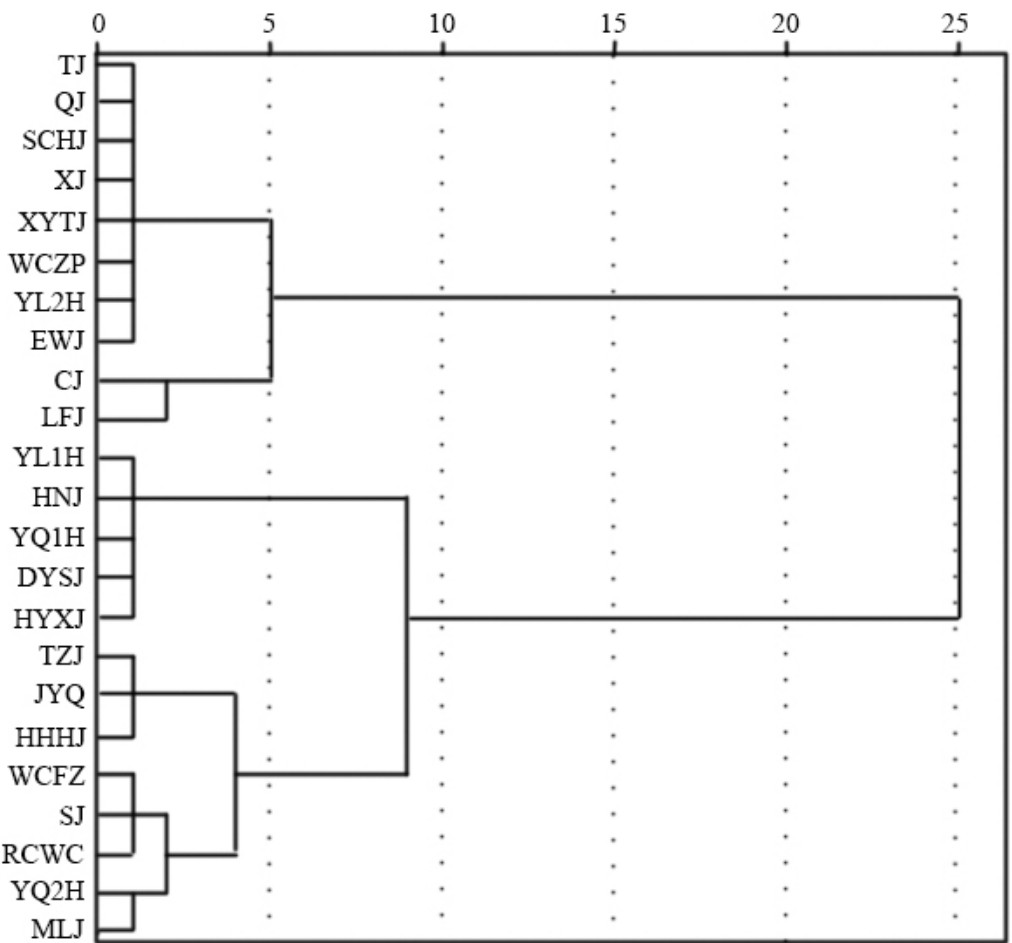

**Figure 10.** Pedigree diagram of cluster analysis based on the physiology and biochemistry of green prickly ash.

The first category included the following eight varieties: TJ, QJ, SCHJ, XJ, XYTJ, WCZP, YL2H, and EWJ. At 0 to −30 °C low-temperature stress, the average REC was 65.91%, which was the third among the four major groups. The SS content was 307.25 μg/mL, the SP content was 189.18 ng/mL, and the Pro content was 6.15 ng/mL, which represented the first, second, and first among the four major groups, respectively. The SOD activity was 111.26 U/mL, POD activity was 321.13 U/L, and CAT activity was 65.72 U/mL, which represented the second, second, and first activities among the four groups, respectively.

The second category included two varieties: CJ and LFJ. At 0 to −30 °C low-temperature stress, the average REC was 62.77%, which represented the fourth among the four groups. The SS content was 305.34 μg/mL, the SP content was 190.28 ng/mL, and the Pro content was 5.50 ng/mL, which represented the second, first, and fourth among the four groups, respectively. SOD activity was 139.83 U/mL, POD activity was 350.30 U/L, and CAT activity was 65.15 U/mL, which were the first, first, and second in the four groups, respectively.

The third category comprised five varieties, including YL1H, HNJ, YQ1H, DYSJ, and HYXJ. At 0 to −30 °C of low-temperature stress, the average REC was 69.01%, the first among the four groups. The SS content was 243.37 μg/mL, SP content was 177.89 ng/mL, and Pro content were 5.67 ng/mL, respectively, which represented the fourth, third, and second among the groups, respectively. SOD activity was 85.38 U/mL, POD activity was 302.12 U/L, and CAT activity was 56.65 U/m, which represented the fourth, third, and fourth levels among the groups, respectively.

The fourth category included the following eight varieties: TZJ, JYQ, HHJ, WCFZ, SJ, RCWC, YQ2H, and MLJ. During 0 to −30 °C of low-temperature stress, the average REC was 66.91%, which was the second among the four groups. The SS content was 303.41 μg/mL, SP content was 176.30 ng/mL, and Pro content was 5.63 ng/mL, which represented the third, fourth, and third highest among the four groups, respectively. The activities of SOD, POD, and CAT were 100.42 U/mL, 294.62 U/L, and 63.77 U/mL, respectively, which represented the third, fourth, and third among the four groups, respectively.

## 4. Discussion

### 4.1. Response of Green Prickly Ash to Low-Temperature Stress

The cell membrane is the first line of defense for plants to resist low-temperature stress and represents the main site where plants sense temperature signals. Consequently, cell membrane stability is closely associated with cold resistance in plants [20]. When a plant senses low temperatures, membrane permeability increases, electrolyte exosmosis occurs, and REC increases. Therefore, the $LT_{50}$ obtained by fitting the logistic equation using conductivity changes can be considered an important indicator of cold resistance in plants [21]. Liu et al. [15] found that the $LT_{50}$ of three red prickly ash varieties was between −17.07 and −20.20 °C, while the $LT_{50}$ of green prickly ash germplasms obtained in this study was between 3.16 and −12.37 °C, which was significantly lower than that of red prickly ash. This indicated that the cold resistance of green prickly ash was weaker than that of red prickly ash, and this difference may be attributed to the origins of these two species: green prickly ash originates from the warm south, whereas red prickly ash originates from the cold north [22]. Furthermore, the $LT_{50}$ of green prickly ash was similar to that of loquat (−2.73 to −4.76 °C) [23], primarily because both are abundant in the southwest and grow in a similar environment. Further analysis showed that the REC of green prickly ash increased rapidly from 0 to −10 °C, and increased gradually from −10 to −30 °C, indicating that the cell membrane of green prickly ash was damaged at 0 to −10 °C and most germplasms were seriously damaged. The $LT_{50}$ values obtained using the conductivity fitting logistic equation further verified this result. The $LT_{50}$ of the 23 green prickly ash germplasms was 3.16 to −12.37 °C, and only 4 green prickly ash germplasms had an $LT_{50}$ lower than −10 °C, indicating that the cold resistance of green prickly ash germplasms differed significantly. Thus, some green prickly ash germplasms have strong cold resistance and can be used as candidates for introduction or as breeding parents for the cultivation of cold-resistant varieties.

Low temperatures cause plant dehydration. Osmotic regulators can regulate the concentration of cellular fluids, reduce the osmotic potential inside and outside the cell, and prevent water loss. Therefore, plants use osmotic regulators to resist low-temperature damage after encountering stress [24]. As low-temperature stress intensifies, osmotic adjustment substances accumulate continuously in peanuts [25], and in citrus plants, the contents of PRO and SS [26] also increase rapidly in the early stage of low-temperature stress. In the present study, the SS and SP contents in green prickly ash branches increased rapidly during the early stage of low-temperature stress and decreased further with decreasing temperature. In the early stages of low-temperature stress, SS and SP increased rapidly to improve resistance to low temperatures. With further elevation of stress beyond the tolerance range, the cell membrane structure is damaged, resulting in a decrease in related enzyme activity and an increase in hydrolase activity, ultimately blocking the synthesis of SS and SP; consequently, the contents of both decrease [27]. Yang et al. [28] found that the

Pro content of red prickly ash during natural overwintering contrasted with the changing trend in SS and SP contents, which initially decreased and then increased with a decrease in temperature and were not synchronized with the increase in the cold resistance of plants. This study found that the Pro content of green prickly ash initially decreased and then increased with a decrease in temperature, which was contrary to the changing trend in SS and SP contents and consistent with the results of previous studies [29]. This shows that there is no distinct correlation between the cold resistance of green prickly ash and Pro accumulation in the body.

Under low-temperature stress, reactive oxygen species in plant cells increase and free radicals accumulate, thus affecting normal activities [30]. Antioxidant enzyme systems remove reactive oxygen free radicals, improve antioxidant enzyme activity, reduce cell damage, and maintain the normal physiological function of cells [31]. Ma et al. [32] found that the SOD activity of red prickly ash initially increased, then decreased, and then increased again with decreasing temperature; POD activity initially increased and then decreased. Liu et al. [33] pointed out that the SOD activity of pomegranate increased with a decrease in temperature but decreased beyond the range of plant regulation. The SOD activity of walnuts initially increased, then decreased, then increased, and then decreased with a decrease in temperature, whereas CAT activity initially increased and then decreased [34]. The results showed that with a decrease in temperature, the SOD, POD, and CAT activities of the green prickly ash decreased. This was consistent with the results reported by Liu et al. [35] for red prickly ash. The antioxidant enzyme activity of plants decreased slightly during the initial stages of low-temperature stress. This may be due to the accumulation of malondialdehyde (MDA) during the early stages of stress, which inhibited the increase in antioxidant enzyme activity. With a continuous decrease in temperature, the antioxidant enzyme systems could be playing a major role leading to increased protective enzyme activity and as the temperature decreased further, it exceeded the tolerance limit of the plant, hence leading to a decrease in the enzyme activity.

### 4.2. Evaluation of Cold Resistance of Green Prickly Ash Germplasm Resources

Many factors affect the cold resistance of plants, and it is difficult to evaluate adaptability to low-temperature stress accurately using a single index. Liu et al. [36] evaluated the cold resistance of six red prickly ash varieties by combining the semi-embolization temperature obtained by hydraulic conductivity using the membership function method and found that the cold resistance of SZT was the strongest among the six varieties. Liu [37] used PCA to evaluate the cold resistance of three red prickly ash varieties and reported that QA1H had the best cold resistance, followed by YJ, whereas DHP had the weakest cold resistance. Liu [38] analyzed the SOD activity, POD activity, MDA content, and SP content of 10 Chinese prickly ash varieties using a subordinate function and found that the cold resistance of DJ and SXHJ was stronger, while that of HYHJ and ZYHJ was weaker. In this study, the cold resistance of green prickly ash germplasm resources was comprehensively evaluated by combining five methods: $LT_{50}$, PCA, correlation analysis, membership function, and cluster analysis. The results of fitting the logistic equation with REC showed that the $LT_{50}$ of HYXJ, DYSJ, and EWJ was above 0 °C and the cold resistance of these varieties was weak, whereas the $LT_{50}$ of TJ, CJ, SCHJ, and WCZP was below −10 °C and the cold resistance of these varieties was strong. According to the results of the PCA, the first category includes 5 pepper varieties with higher cold-resistance scores, including CJ, LFJ and XYTJ; the second category includes 13 pepper varieties with appropriate scores, including YQ2H, WCFZ, and TZJ; and the third category includes 5 pepper varieties with lower scores, including HYXJ, HNJ, and DYSJ. There may be great differences in the cold-resistance evaluation of green prickly ash with a single index, correlation analysis found. Furthermore, a significant negative correlation was observed between SOD activity and REC, with a correlation coefficient of −0.56, and a significant positive correlation between REC and PRO and between SOD and CAT activity, with correlation coefficients of 0.5 and 0.43, respectively. The membership function analysis of seven cold resistance physiological

and biochemical parameters of the 23 green prickly ash germplasms showed that the comprehensive scores of LFJ, CJ, and EWJ were the highest and their cold resistance was the strongest, while the comprehensive scores of DYSJ, YQ1H, and HYXJ were the lowest and their cold resistance was the weakest. The results of the cluster analysis showed that when the Euclidean distance was 5, the 23 green prickly ash germplasms were divided into four groups, among which CJ and LFJ were the first and had the strongest cold resistance. Five varieties, including HYXJ, DYSJ, and HNJ, were in the first group, and their cold resistance was the weakest. The first three of the five evaluation methods included CJ, and the last three included DYSJ and HYXJ. In summary, the top three prickly ash varieties of cold resistance in the five evaluation methods include CJ, and the last three pepper varieties include DYSJ and HYXJ; the top three prickly ash varieties of membership function method and cluster analysis method included LFJ, and the low temperature $LT_{50}$ of LFJ was also low. Therefore, CJ and LFJ can be used as cold-resistant varieties of prickly ash, and DYSJ and HYXJ can be used as sensitive varieties of prickly ash.

**5. Conclusions**

Under low-temperature stress, the physiological responses of green prickly ash germplasm differed significantly. The $LT_{50}$ of different green prickly ash germplasms was between 3.16 and $-12.37\,°C$. As the temperature decreased, the REC of green prickly ash increased gradually. A rapid increase in REC was observed in the early stage of stress, and a gradual decrease occurred with an increase in stress. The content of SS initially increased and then decreased; the content of SP initially increased, then decreased, and then increased; and the content of Pro initially decreased and then increased. The SOD activity decreased, POD activity decreased, and CAT activity increased with cold stress. In this study, to evaluate the cold resistance of plants comprehensively and accurately using various indexes and overcome the limitation of single index identification, we combined five methods: conductivity for $LT_{50}$, PCA, correlation analysis, membership function method, and cluster analysis. Overall, the varieties CJ and LFJ were considered cold-resistant while DYSJ and HYXJ were cold-sensitive. The findings presented here provide theoretical guidance and technical support for the screening of cold-resistant germplasms of green prickly ash and will aid in the introduction of these trees in northern China.

**Supplementary Materials:** The following supporting information can be downloaded at: https://www.mdpi.com/article/10.3390/horticulturae9111242/s1. Table S1. Changes of REC content in green prickly ash germplasm from different origins under different low temperature stress. Table S2. Changes of SS content in green prickly ash germplasm from different origins under different low temperature stress. Table S3. Changes of SP content in green prickly ash germplasm from different origins under different low temperature stress. Table S4. Changes of PRO content in green prickly ash germplasm from different origins under different low temperature stress. Table S5. Changes of SOD content in green prickly ash germplasm from different origins under different low temperature stress. Table S6. Changes of POD content in green prickly ash germplasm from different origins under different low temperature stress. Table S7. Changes of CAT content in green prickly ash germplasm from different origins under different low temperature stress.

**Author Contributions:** Q.L. and Z.C. designed the study. H.F. and X.D. conducted the experiments. X.C. and S.B. analyzed the data. Y.R., K.H. and Z.C. developed the new methods. L.S. wrote the manuscript. All authors have read and agreed to the published version of the manuscript.

**Funding:** This study was supported by the Scientific and Technological Research Program of the Chongqing Municipal Education Commission (KJQN202201305), Natural Science Foundation of Chongqing (cstc2019jcyj-msxmx0803), Projects for Innovative Research Groups of Chongqing Universities (CXQT21028), and the Chongqing Talent Program for Zexiong Chen. The funders played no role in the study design, data collection, data analysis, data interpretation, or manuscript writing.

**Data Availability Statement:** All data supporting the findings of this study have been included in this article.

**Acknowledgments:** We would like to express our gratitude to the Chongqing University of Arts and Sciences for providing the green prickly ash materials for this study.

**Conflicts of Interest:** The authors declare no conflict of interest.

**Abbreviations**

| | |
|---|---|
| REC | Relative conductivity |
| LT50 | Lethal dose-50 temperature |
| SS | Soluble sugar |
| SP | Soluble protein |
| PRO | Free protein |
| SOD | Superoxide dismutase |
| POD | Peroxidase |
| CAT | Catalase |

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
