# Peer review of "Differences in Physiological Characteristics of Green Prickly Ash Germplasm Resources in Response to Low-Temperature Stress"

_horticulturae, doi:10.3390/horticulturae9111242_

Round 1

Reviewer 1 Report (Previous Reviewer 1)

Comments and Suggestions for Authors

Authors have studied the physiological responses of green prickly ash lines collected across China to low temperature stress. The major concerns of the study are:

1. The cold treatment was performed using 1 year old branches of the stress and the cold exposure was performed in a refrigerator instead of a controlled growth chamber or incubator where the light and humidity conditions could be regulated. Hence the observations made by the authors might not truly reflect the responses shown by these different varieties.

2. The figures of the data are so crowded with the significance letters that it was hard to understand them. I feel that the authors should come up with an easier way to name the lines that could be followed easily while reading the manuscript.

3. In the methods section, the formula used to determine LT50 was not clear as the letters mentioned in the formula were not described.

3. Abbreviations like REC, SS, SP, SOD etc. were used all throughout the manuscript but it would be good to elaborate them at the first time they are being used in the text.

I feel that the manuscript with the current data would not be enough to attract the interest of readers and hence could not be accepted for publication.

Comments on the Quality of English Language

The manuscript needs a moderate correction for the english language.

Author Response

Reviewer 2 Report (Previous Reviewer 2)

Comments and Suggestions for Authors

N/A

Comments on the Quality of English Language

N/A

Author Response

Reviewer 3 Report (New Reviewer)

Comments and Suggestions for Authors

Dear authors

The following modifications are required

Abstract

ü  In general, this section is poorly written. It is written simply. This section should include. As a result, this section should be improved.

ü  Before describing the goal, the authors must define the issue in a single line and explain why they chose this approach to study this review.

ü  No information about the type of experimental design and its component is available in this manuscript.

ü  Some quantitative data should be added

ü  In the final line of the abstract, the authors should present a decisive conclusion derived from the research and provide a single line of future prospects.

Keywords

ü  The content of keywords did not reflect the content of this manuscript and the words used for forming the title should not be used as the keywords. So, the structure of keywords should be changed.

Introduction

ü  Detail information about the impacts of low temperature on the biochemical parameters

ü  The authors should give some lines about the knowledge gap which their reviews have covered along with the hypothesis statement

ü  Also, the authors should provide a novelty statement at the end. What new things authors have done or correlated in this research compared to old ones?

ü  The general and specific aim should be specified

Materials and Methods

ü  No information about the type of experimental design and its component is available in this manuscript.

ü  The authors should mention the method of comparison between the means in the section of statistical data analysis

ü  The authors should write the number of replications and the number of plants per replications.

ü  The authors should mention the type of tissue used for the biochemical tests.

ü  The type of soil (Silt, sand….etc) should be mentioned

ü  All abbreviations should be written in full name

Results and discussion

ü  In general, the figures are not presented clearly.

ü  The status of significant of each studied parameter should be mentioned at the beginning of the text

ü  The authors should mention the scored data when they explain the maximum and minimum of the studied traits.

ü  All captions should be improved, showing the contents of tables and figures

ü  The method of the comparison of means (LSD, Duncan, Tukey, Dunnett) should be included

ü  A PCA plot should be created to better understand the tolerance index.

ü  The correlation and probability values of each pair of traits should be mentioned in the text

ü  The discussion is weak. The authors should interpret all results obtained in this study by adding some information about the results obtained in their study. The authors should explain how all of the findings from this study relate to their own findings. The authors should explain the impact of cold on the physiochemical traits by adding the mechanism of affecting

Conclusion

ü  The authors should summarize the most significant findings because they have written this section in an easy-to-read manner.

ü  Future works about this research should also include additional works

Comments on the Quality of English Language

Extensive correction is needed

Author Response

Reviewer 4 Report (New Reviewer)

Comments and Suggestions for Authors

Dear Authors, 

My opinion is below:

My findings are:

The topic is correct and worth dealing with, the methods used are good.

The article is formally incorrect. The proofreadings were left inside, which makes it very difficult to properly review the manuscript.

The reference literature is good, as is the Introduction chapter, but the literature results of the last 5 years are currently omitted, which I recommend to be included in the manuscript, thus the Introduction can be significantly expanded.

Material and Methods are fine.

In the results, figures 1-7 are difficult to see, their quality is not good, it would be good to improve their quality or make the values more visible, because this is not appropriate.

Conclusion: what has been described is good, but the chapter should be expanded with long-term plans, why it is good for agriculture in our current world, etc.

Round 2

Reviewer 1 Report (Previous Reviewer 1)

Comments and Suggestions for Authors

Comments on the Quality of English Language

Moderate editing of english language is required. Authors need to check the tense carefully and follow it across the entire text.

Author Response

Please refer to the attachment

Reviewer 3 Report (New Reviewer)

Comments and Suggestions for Authors

The authors have been addressed all comments

Comments on the Quality of English Language

moderate correction is needed

Author Response

Please refer to the attachment

Reviewer 4 Report (New Reviewer)

Comments and Suggestions for Authors

Dear Authors,

you did a good job revising the manuscript. The Abstract is too long - after shortening it (as MDPI recommends), the manuscript can be published.

Author Response

Please refer to the attachment

This manuscript is a resubmission of an earlier submission. The following is a list of the peer review reports and author responses from that submission.

Round 1

Reviewer 1 Report

Comments and Suggestions for Authors

Authors have studied the effect of cold stress in 23 green prickly ash varieties to identify the tolerant ones to be used in Northern region of china. 

The major concern/flaw is with the method used to screen the varieties with the stress treatment. Authors have used the branches from the 3 year old trees and subjected them to cold stress in a refrigerator. The conditions provided in this study does not reflect/mimic the cold stress the plant would experience. Authors should have used seedlings of the varieties to study the effect of cold stress in a growth chamber with controlled growth conditions.

The details of the cold stress treatment provided in the methods section are also not clear. They have used acronyms at most of the places and have not mentioned them clearly. For example, they have measured SS, SP and AEC but have nowhere mentioned what do they stand for. Hence the study lacks the basic scientific approach and hence could not be accepted for the publication.

Comments on the Quality of English Language

Authors have written the manuscript in a decent readable manner. However, it needs moderate language corrections.

Reviewer 2 Report

Comments and Suggestions for Authors

There are some major flaws:

1. the methods or different assays need to explained in detail

2. the experiment need a control to show adverse effects in cold stress and another control which shows no or minimal effect. I am not sure if this was clear in the manuscript.

3. The results in the present form will not interest readers. The authors really need to make the results stand out. Numerical values in a table format doesn't form a good foundation of judging the results or experimental design. Please use proper test of significance like ANOVA or other along with graphical presentation.

4. The conclusion section needs to highlight the message of "what was learnt from this study and how this information will be used in future for both fundamental and applied research". What's the broad goal that authors want to achieve by this study? This is very important to drive the message home.

Comments on the Quality of English Language

Please proof-read multiple times for grammar and typos